# Backtracking: Improved methods for identifying the source of a deliberate release of *Bacillus anthracis* from the temporal and spatial distribution of cases

**Joseph Shingleton**, **David Mustard**, **Steven Dyke**, **Hannah Williams**, **Emma Bennett**, **Thomas Finnie** *

Data, Analytics and Surveillance; UK Health Security Agency; Porton Down, United Kingdom

\* Thomas.Finnie@ukhsa.gov.uk

**Data Availability Statement:** Access to the code used within this paper is available to bona fide researchers via academic partnership with UKHSA.

## Abstract

Reverse epidemiology is a mathematical modelling tool used to ascertain information about the source of a pathogen, given the spatial and temporal distribution of cases, hospitalisations and deaths. In the context of a deliberately released pathogen, such as *Bacillus anthracis* (the disease-causing organism of anthrax), this can allow responders to quickly identify the location and timing of the release, as well as other factors such as the strength of the release, and the realized wind speed and direction at release. These estimates can then be used to parameterise a predictive mechanistic model, allowing for estimation of the potential scale of the release, and to optimise the distribution of prophylaxis.

In this paper we present two novel approaches to reverse epidemiology, and demonstrate their utility in responding to a simulated deliberate release of *B. anthracis* in ten locations in the UK and compare these to the standard grid-search approach. The two methods —a modified MCMC and a Recurrent Convolutional Neural Network—are able to identify the source location and timing of the release with significantly better accuracy compared to the grid-search approach. Further, the neural network method is able to do inference on new data significantly quicker than either the grid-search or novel MCMC methods, allowing for rapid deployment in time-sensitive outbreaks.

## Author summary

In this paper we demonstrate three methods for estimating the source location and timing of a deliberate release of *Bacillus anthracis* based on the temporal and spatial distribution of cases. Two of our proposed methods, a modified MCMC approach and a neural network based approach, provide significant improvements over previous methods by directly addressing the problematic parameter-likelihood surface, and, in the case of the neural network approach, addressing the slow deployment speeds of existing methods.

More information on partnership with the agency is available at https://research.ukhsa.gov.uk/partnerships/ and initial contact may be made via ResearchSupport@ukhsa.gov.uk.

**Funding:** JS, DM, SD, HW EB and TF were funded via United Kingdom Department of Health and Social Care (https://www.gov.uk/government/organisations/department-of-health-and-social-care) grant-in-aid funding to UKHSA. The funders had no role in study design, data collection and analysis, decision to publish, or preparation of the manuscript.

**Competing interests:** The authors declare that they have no competing interests.

Our results represent a major step forward in the accuracy and speed of epidemiological back-calculation.

## Introduction

A core part of the public health response to the deliberate or accidental release of a wind or environmentally dispersed pathogen is in identifying the location and timing of the release [1, 2]. During such an event, it is likely that the only data streams available to responders will be incomplete line-lists detailing the location of individuals when symptom onset occurs, as well as the time of symptom onset, and the timing of any subsequent hospitalisations or deaths [1].

The process of using line-list style data to infer information about the source of the pathogen is called *reverse-epidemiology*. Typically, this process starts by using a forward model which predicts the geographic and/or temporal distribution of cases for an initial set of joint disease and transportation model parameters. Example models are discussed extensively in [3], in which various forward models for *Bacillus anthracis* dispersion and symptom progression are evaluated. The chosen forward model with the initial parameter set is then optimised to find the most likely set of parameters, such as source location, given the observed outbreak (from the early cases from line list). Once a new/updated parameter set is identified, within given confidence limits, the forward model with these fitted parameters is used to estimate the timing and location of any subsequent cases, allowing for the efficient distribution of prophylactic antibiotics [2, 4].

The forward model used in this paper comprises of two parts. Firstly, Briggs' dispersion model [5] is employed to simulate the dispersion of *B. anthracis* organisms across a two-dimensional plane representing ground level. This is parameterised by wind speed, wind direction, and strength of source; with other factors such as organism decay rate and individual breathing rate fixed at pre-determined values. Using this model we can estimate the dose received at each planar location. Secondly, by combining this with a UK population density map [6] it allows us to estimate the number of individuals exposed to the pathogen at each planar location. A within host disease model [7] is then used to estimate the incubation period for each person infected by the pathogen, as well as models for the time between symptom onset and potential hospitalisation and death [8]. The combination of these two models provides a reasonable simulation of the spatial and temporal distribution of cases, hospitalisations and deaths following a *B. anthracis* release in a given location.

A common approach to reverse-epidemiology is to derive some likelihood function related to the forward model [2, 3], then use a method which searches the parameter space to find the set of parameters which maximises the likelihood of a given set of observations. However, care is needed when taking this approach. Depending on the complexity of the forward model the parameter-likelihood surface can often be extremely complicated—featuring many local maxima and steep gradients. As such, a simple gradient descent approach is often unsuitable and will tend to given erroneous results. This problem is worsened with increased model complexity.

A common alternative to direct parameter optimisation is using Markov Chain Monte Carlo (MCMC) stimulation to form a Bayesian posterior distribution of the parameters, (see for example Gilks et al. [9]). However, this approach also can have difficulties when the posterior distribution is multimodal, particular when the local maxima are separated by "valleys" of very low probability. Addressing these difficulties is a major area of active research. Medina-Aguayo and Christen [10] describe parallel tempering as "the state-of-the-art method for dealing with multimodality" but believe that the computational costs of the high number of

tempered distribution and iterations normally needed can be prohibitive, as would be the case here. As an alternative, they extend the t-walk MCMC method of Christen and Fox [11] by adding a fifth move type to the four original move types. This "penalised proposal" move is intended to counteract the common problem of the MCMC not fully exploring the posterior due to becoming entrenched near a local maxima which is not the global maxima. Another aspect of t-walks is that at each step only a random selection of the parameters are updated. The MCMC method presented in the current paper shares with the t-walk algorithm multiple move types with the type used at a particular step being chosen at random. It also has a stage where an individual parameter rather than all parameters are updated during a step. The method also has a trial period for certain proposal acceptances which can be viewed as short-lived parallel chains.

As an alternative to the use of likelihoods in estimation of the parameters of dynamical systems, several authors have considered the use of neural networks, in particular long short term memory (LSTM) based recurrent neural networks. Lee [12] applied such a model to temporal financial data, finding that it gave a similar level of accuracy as maximum-likelihood estimation but with a greatly reduced computation time. Rudi et al. [13] compared the performance of dense and convolutional neural networks in predicting the parameters of a established temporal model of the membrane potential of biological neurons. They concluded that the convolutional neural network gave the better results when the data was noisy. A network formed by the combination of a LSTM network and a fully connected network was applied to the estimation of parameters of a Lévy noise-driven stochastic differential equation system by Wang et al. [14]. They found it gave higher accuracy and much quicker results than maximum likelihood-based approaches. They did not relate the studied system to any physical process and suggested that the approach is easily generalizable to other stochastic systems. Convolutional neural networks were applied by Lenzi et al. [15] to two-dimensional spatial max-stable problems, including the modelling of USA temperature data, which are generally considered too computationally challenging for traditional maximum-likelihood estimation. They report considerable improvements in both computational time and accuracy over a pairwise-likelihood approach. Within the field of epidemiology, Raissi et al. [16] investigated the use of a Physics informed Deep Learning model, (a type of neuron network associated with differential equations), to fit the parameters of a simple susceptible, infected and recovered (SIR) compartmental model to time series data from the well studied English boarding school influenza outbreak of 1978 [17]. Compared to the above the neural network model outlined in the current paper has been developed to be applied to spatio-temporal data rather than purely spatial or temporal data.

In this paper we present three approaches. First, we introduce the standard grid-search technique which searches the entire parameter space for the parameter set with the maximum likelihood, given an observed outbreak. The second approach, a modified MCMC optimizer, also relies on the likelihood function but implements novel strategies for avoiding the local maxima and steep gradients in the parameter-likelihood surface. Finally, we introduce a Recurrent Convolutional neural network (RCNN) approach which does not explicitly rely on the likelihood function. In this approach we randomly sample the parameter space a large number of times, and use the forward model to simulate an outbreak from each parameter set. The RCNN model is then trained to estimate the parameter set used to produce a given input.

## Materials and methods

### Forward models

A key part of the reverse-epidemiology process is the forward model. This can be described as some function $F(x, y, t; \Theta) \mapsto C(x, y, t)$ which provides the number of cases at a given location

and time, dependant on some set of parameters, $\Theta = \{p_x, p_y, t, S, U_s, W_d\}$, specific to the release. See below for an explanation of these parameters.

For this paper, we consider the forward model to be a combination of a dispersion model, describing the transportation of the disease particles from the source (via atmospheric dispersion) and ultimately yielding the dose received at each $x$, $y$ location; and a disease course model which describes the progression of the disease from exposure, through symptom onset, and to possible hospitalisation and death.

The dispersion model used is based on a Brigg's dispersion model [5], similar to that described in [3]. For a given planar location, the model provides an expected pathogen (inhaled) dose given by:

$$\frac{C_1}{U \sigma_x \sigma_z} S \exp\left( -\frac{1}{2} \left( \frac{y^2}{\sigma_x^2} + \frac{z^2}{\sigma_z^2} - C_2^2 \right) - C_2 C_3 \right) \Phi(C_3 - C_2), \tag{1}$$

where

$$C_1 = \frac{B}{\pi}, \tag{2}$$

$$C_2 = \frac{\sigma_x k}{U}, \tag{3}$$

$$C_3 = \frac{x}{\sigma_x}, \tag{4}$$

and

$$\Phi(x) = \frac{1}{\sqrt{2\pi}} \int_{-\infty}^{x} \exp\left( -\frac{t^2}{2} \right) dt \tag{5}$$

is the cumulative density function of the standard normal distribution. The remaining notation is given in Table 1.

The dispersion variances are given by:

$$\sigma_x = \frac{a\hat{x}}{(1 + b\hat{x})^c} \tag{6}$$

**Table 1. Notation used in statement of model.**

| Symbol | Description | Notes |
|--------|-------------|-------|
| $x$, $y$ | geographical coordinates | The coordinate system is taken as such that the release occurs at the origin and the wind blows in the direction of the $x$-axis |
| $z$ | Height of release | Set to be 2 m |
| $\sigma_x$ | Dispersion variance ($x$-direction) | Discussed below |
| $\sigma_z$ | Dispersion variance ($z$-direction) | Discussed below |
| $U$ | Wind speed | Taken to be a positive constant |
| $S$ | Source strength | |
| $B$ | Breathing rate | Volume of air inhaled in unit time. |
| $k$ | Decay rate | Rate at which pathogen becomes unviable. |

**Table 2. Parameter values used to calculate dispersion variance.**

| Day/Night | Wind Speed | $a$ | $b$ | $c$ | $a_z$ | $b_z$ | $c_z$ |
|---|---|---|---|---|---|---|---|
| Day | Low | 0.32 | 0.0004 | 0.5 | 0.24 | 0.001 | −0.5 |
| Day | Medium | 0.22 | 0.0004 | 0.5 | 0.2 | 0 | 0 |
| Day | High | 0.16 | 0.0004 | 0.5 | 0.14 | 0.0003 | 0.5 |
| Night | Medium or High | 0.16 | 0.0004 | 0.5 | 0.14 | 0.0003 | 0.5 |
| Night | Low | 0.11 | 0.0004 | 0.5 | 0.08 | 0.00015 | 0.5 |

and

$$\sigma_z = \frac{a_z \hat{x}}{(1 + b_z \hat{x})^{c_z}}, \tag{7}$$

where

$$\hat{x} = \max(x, 1) \tag{8}$$

and $a$,$b$,$c$,$a_z$,$b_z$ and $c_z$ are independent of location but depend on whether it is day (between 7am and 7pm) or night, and whether the wind speed, $U$, is low ($U \leq 4$ ms$^{-1}$), medium ($4$ ms$^{-1}$ $< U \leq 6$ ms$^{-1}$) or high ($U > 6$ ms$^{-1}$). These values are given in Table 2.

The first stage of dose-dependent disease course, the time between exposure and symptom onset or incubation period is modelled using Wilkening's least square fit to a competing risk in-host model with log-normal distributed bacterial growth [7].

The other two stages of the disease course, the lag time between symptom onset and hospitalisation, and between hospitalisation and death are modelled using the models of Holty et al. [8] These models are derived from fitting data from historical outbreaks to log-normal distributions. The standard deviation $\sigma$ and natural log of the scale ln $m$ of these fits are shown in Table 3.

The forward models produce a simulated line-list of geographically and temporally distributed cases, based on five variable source parameters: location, ($p_x$, $p_y$); release time, $t$; source strength (as the number of organisms released), $S$; wind speed (in meters per second), $U_s$; and wind direction, $W_d$, as well as a number of fixed parameters (see Table A in S1 Supporting Information for full list of fixed and variable parameters). Simulated line-lists also have an associated censor date—that is the date on which the line-list is produced. No cases, hospitalisations or deaths are recorded after the censor date, even if such events occur in the simulation.

## Likelihood function

This section describes the likelihood function used in the grid-search and MCMC methods. We employ the same likelihood function as described in [3]. This function has the following

**Table 3. Parameter values used in the two-parameter log-normal distributions used to model within host dynamics, as described in [8].**

| Time period | ln $m$ | $\sigma$ |
|---|---|---|
| Symptom onset to hospitalisation | 1.3 | 0.5 |
| Hospitalisation to death | −0.3 | 0.9 |
| Symptom onset to death | 1.567 | 0.45 |

equation:

$$\prod_{i=1}^{cases}\left(\frac{a_i f_i}{F_i}\right)\prod_{i=1}^{non-cases}(1-a_i F_i) \tag{9}$$

where *cases* are individuals who are symptomatic by the censor date, and *non- cases* are individuals who are not symptomatic by the censor date [3].

In above notation $a_i$ is the attack rate for case $i$ (based on the dose which was deposited at the location of case $i$), $f_i$ is the probability density function (PDF) of the disease symptom onset distribution evaluated at the symptom onset time for case $i$, $F_i$ is the cumulative distribution (CDF) of the disease distribution, evaluated at the censor time, for case $i$. The calculation of this likelihood function first of all requires an application of the dispersion model to determine the dose inhaled at each location. We opt for returning the log of the likelihood from this function. If at any point any component is calculated as zero the calculation is abandoned and a value of $-\infty$ is returned as the log-likelihood.

In Eq (9) the first factor is the probability that each case occurred when it did, given that it occurred by the censor time. The second is the probability that each non-case had either not been infected by the censor time or, if infected, was not showing symptoms by this time. Thus the likelihood reflects the probability that current cases occurred at the time that they occurred, and that non-cases have not occurred by the censor date.

For each individual, the PDF and CDF are weighted by the attack rate, i.e. the probability of being infected, for the corresponding location. The inclusion of the attack rate means that a large number of non-cases occurring in low dose locations has no adverse effect on the likelihood (i.e. a large number of non-cases at low dose location is to be expected). Conversely a large number of cases at these low dose location would greatly reduce the likelihood.

The likelihood based approaches discussed below have two major drawbacks, both illustrated in Fig 1, which sketches a possible scenario where the estimated wind direction may become trapped in a plausible region due to an adjacent area with high population density and zero observed cases.

Parts of parameter space with higher likelihood can often be surrounded by areas with log-likelihood approaching negative infinity. This is often because that part of parameter space corresponds to cases occurring in areas with no population, or that the resulting solution can not account for the lack of cases occurring in a region with high population density. In Fig 1 we see that while the current solution, $\hat{W}_d^1$, has a relatively high likelihood, it is separated from the true solution, $W_d$, by a region with log-likelihood approaching negative infinity. Because humans tend to aggregate in space, this is a common problem when searching through the space of spatial parameters ($p_x$, $p_y$, $W_d$), and does not generally occur with the time parameter, $t$, which is much more dependent on the smooth incubation period distribution.

The likelihood based approaches also implicitly rely on a degree of smoothness in the parameter-likelihood surface. In Fig 1b we see a proposed solution, $\hat{W}_d^2$, close to the true solution, $W_d$. If the step size used is too large, then $\hat{W}_d^2$ will be misidentified as the solution with the highest log-likelihood.

In the example in Fig 1 we consider how regions with negative-infinity log-likelihood can occur when varying a parameter in one dimension. More commonly, however, these regions arise when we vary both the wind direction, $W_d$, and the source location, ($p_x$, $p_y$). Fig 2 illustrates how the space of possible parameters, rather than being continuous, is generally divided into distinct plausible regions. Frequently, the plausible region containing the true solution is accompanied by a second plausible region directly opposite, with the wind direction rotated by 180°. There may additionally be (many) other islands of plausibility.

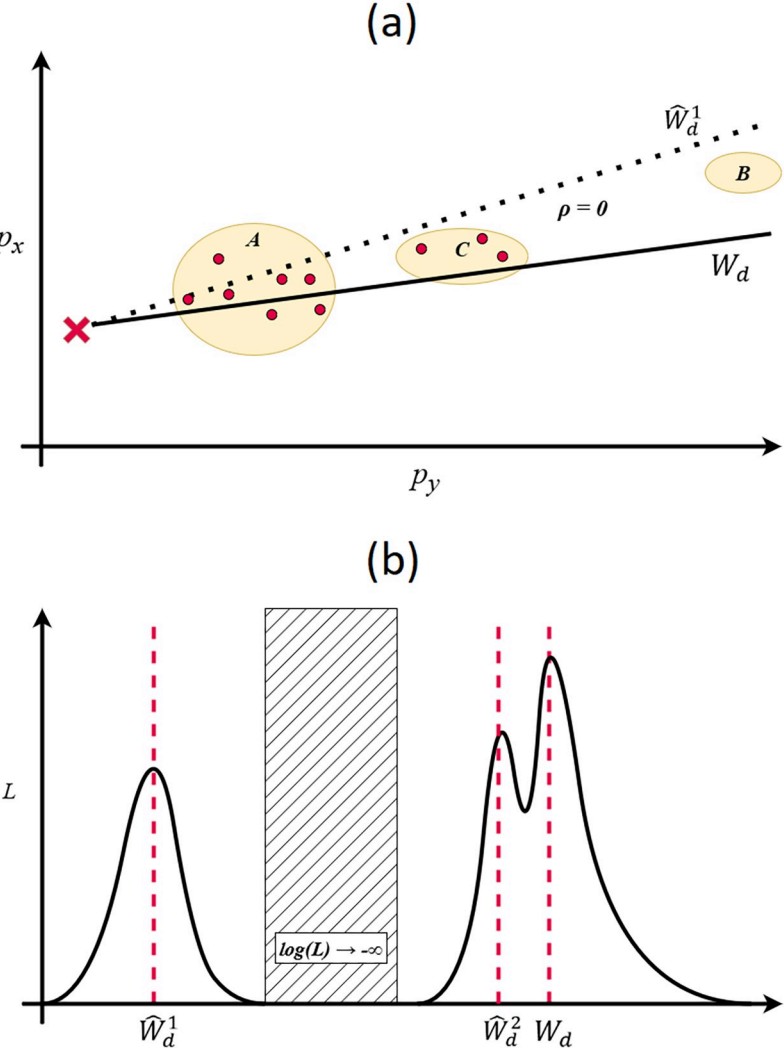

**Fig 1.** (a) shows a possible scenario in which wind direction is being varied for a given $(p_x, p_y)$ (illustrated by the red cross). Regions $A$, $B$ and $C$ have population density $\rho \gg 0$, all other locations have $\rho = 0$. The current prediction, $\hat{W}_d^1$ (dashed line), points toward region $A$ and passes near region $C$. The true parameter value, $W_d$, passes through regions $A$ and $C$. Neither solution passes through region $B$. Cases, illustrated by the red dots, occur in regions $A$ and $C$, but do not occur in region $B$. As such, both $\hat{W}_d^1$ and $W_d$ are plausible solutions. However, due to the absence of any cases in the region, any solution pointing through region $B$ would have $\log(L) \to -\infty$. This is illustrated in (b) which sketches the likelihood of different values of $W_d$. For most gradient descent based methods it would not be possible for $\hat{W}_d^1$ to be improved upon as it would require the estimate to pass through a region where $\log(L) \to -\infty$. We also illustrate how the smoothness assumption limits the utility of such approaches. Depending on the step size used in such an approach, it is possible that a gradient descent method may misidentify the local maxima at $\hat{W}_d^2$ as the true solution.

## Inference method 1: Grid-Search

The first, and perhaps simplest, approach to optimising the likelihood function is to use a grid-search method to assess the function across all feasible parameter space. The log-likelihood function is evaluated on a uniform parameter grid, with the strength parameter, $S$, being considered in log space. The parameter estimate is simply taken as the set of parameter values with the greatest log-likelihood, given the observed case distribution, amongst the sampled values.

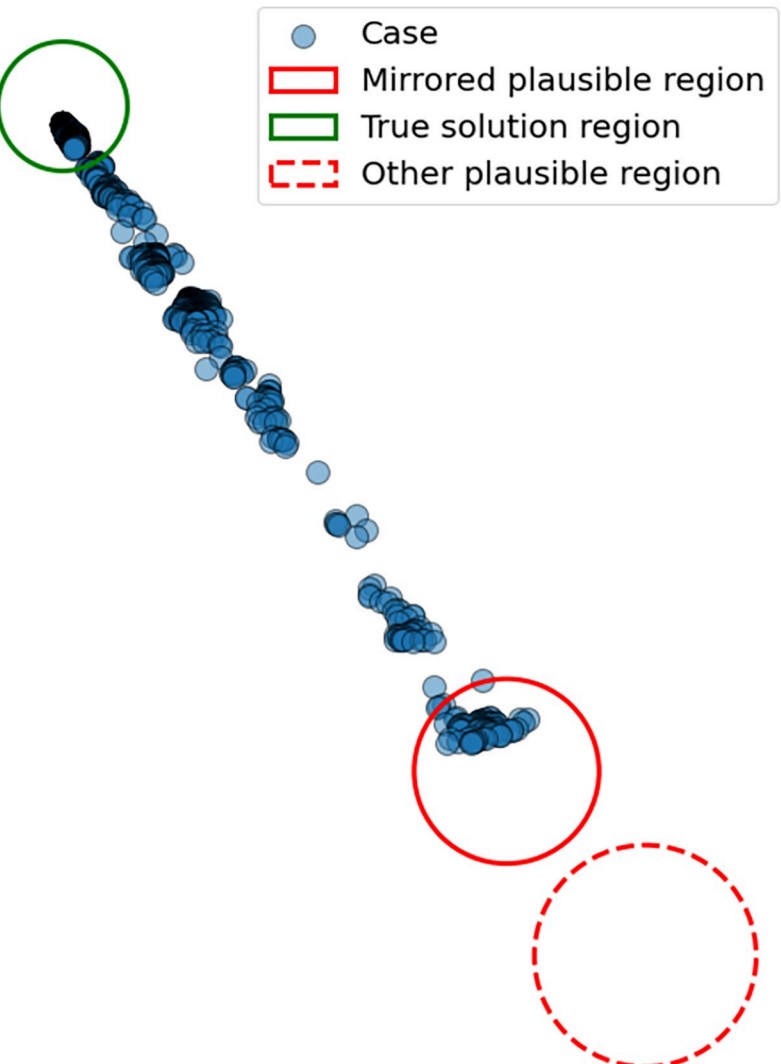

**Fig 2. Plausible solution regions for a given spatial distribution of cases (blue points).** The true source location is somewhere in the green region, however the MCMC method will occasionally arrive at a solution in a region diametrically opposed to the true solution, indicated here by the solid red circle. Other plausible regions may also exist, as indicated by the red dashed region.

Details of the parameter ranges and step sizes used in this approach are given in section A of S1 Supporting Information.

This approach is not limited by the constraints of the parameter-likelihood surface but it will only find the global maxima in the highly unlikely situation that this falls at one of the grid points searched. Precise location of this maxima would generally require a second step gradient decent or adaptive grid-search based on the maxima found. We have not reported on such a multi-step process in this paper as they are limited by the effectiveness of the initial search.

### Inference method 2: MCMC

The second approach to the problem involves the application of a standard MCMC optimisation method [9], an approach previously applied to a similar problem in Legrand et. al. [2].

However, we have modified the method to address some of the problems with the likelihood function described above. In this method we randomly select an initial starting point within the the parameter space. Thereafter, the solution advances by choosing a new point based on a sample drawn from a random distribution centred on the current point. An acceptance test (which is based on the likelihood function) is then applied to decide whether or not this new point should be accepted. If accepted, the new point becomes the current point—if not, the process is repeated for the unchanged current point.

The standard MCMC approach of gradual adjustments to the current solution is not generally able to move from one plausible region to another due to regions with negative log-likelihood dividing them. Thus, if the initial proposed solution is in an incorrect region the solution chain will be trapped on this region and away from the true solution. In the case illustrated in Fig 2 this leads to a 180˚ error in the wind direction, and a corresponding misidentification of the source location.

By examining the orientation of the case data it is possible to make an initial estimate of the wind direction. Of course, it might not be similarly possible to decide if the release is travelling up or down that path—.i.e the wind direction might be the given orientation, or that orientation rotated by 180˚. However, imposing this estimated wind direction on the otherwise randomly chosen initial point of the MCMC procedure means the solution is starting off from either the true solution region or its mirror image region.

Once the calculation of the solution is underway the standard MCMC method of selecting a new potential point is replaced by a *Rotate* method at randomly selected steps. This methods rotates the source location by 180˚ about the center of mass of cases, along with a 180˚ rotation of the current wind-direction estimation. The source location is then randomly moved towards or away from the the center of mass of cases, along the line joining the new position to the center of mass. In addition to the linear movement of the source location there is a corresponding adjustment made to the source strength (increased if the distance between source and center of mass is increased, and decreased otherwise).

Upon completion of this rotation function we apply the standard MCMC acceptance test to determine whether the new point should be accepted as the new current solution. If the rotated solution is not accepted, but it is at least a plausible solution, (i.e. one with finite log-likelihood), rotating back is suppressed for a fixed number of iterations. This allows the standard MCMC to improve on the rotated solution, before a final decision is made on whether the solution should be accepted.

The above procedure has been adopted to address the specific problem of disjoint regions of plausible solution space. This process improves the method's ability to identify the location of the source, however a further modification was required to improve performance on other parameters.

In the standard MCMC iteration, the new solution point is found by randomly varying all five parameters simultaneously. An improved final solution was found if, at the end of a number of iterations (either standard MCMC iterations or rotate iterations), a further number of iterations is performed where each of the five parameters is changed one at time. After each parameter has been varied individually to produce a potential new solution, a standard MCMC acceptance test is performed to decide if this new solution should be accepted.

## Inference method 3: Recurrent-Convolutional Neural Network (RCNN)

The first two approaches discussed in this paper are limited by both the behaviour of the likelihood landscape and the complexity of the forward model. The approaches also require the

forward model to be evaluated a very large number of times during inference, which leads to long calculation times. Increasing the complexity of the forward model exacerbates this problem.

In this section we propose an alternative solution which does not rely on an explicit likelihood function. This method utilises neural networks to learn the inverse of the forward model. The network can be trained on thousands of simulated outbreaks prior to any need for inference, and run very quickly when source-term inference is required. Further, the Neural Network model is able to easily leverage more information from the line-list data by incorporating the timing of hospitalisations and deaths—data which is currently unused in the MCMC and grid-search approaches.

In this approach, we treat each simulated outbreak as a single observation when training the neural network. The forward model parameter set, $\Theta = p_x, p_y, t, S, U_s, W_d$, used to produce each outbreak is used as the response variable for the model. Each observation in the model consists of three inputs; $X_1 \in \mathbb{R}^{N_t \times N_x \times N_y \times 3}$ describing the relative spatial and temporal distribution of cases, hospitalisations and deaths; $X_2 \in \mathbb{R}^{1 \times 9}$ providing a positional reference for $X_1$, along with information about the scale of the outbreak; and $X_3 \in \mathbb{R}^{N_x \times N_y}$ describing the local population distribution in the geographic region spanned by the case data, using data provided by the UK Health and Safety Executive [6]. A detailed description of how these inputs are constructed is given in the supplement (section B of S1 Supporting Information).

The response variable, $\Theta$, undergoes a transformation prior to use in the model. The wind speed, $U_s$, and wind direction, $W_d$, are transformed into cartesian coordinates such that $\eta_1 = U_s \sin W_d$ and $\eta_2 = U_s \cos W_d$. This removes the discontinuity in the wind direction as $W_d$ approaches 0˚ or 360˚, which would otherwise lead to unnecessary penalisation in the loss function. To improve model efficiency we standardised the target variable so that each parameter has mean of 0 and standard deviation of 1.

Recurrent-Convolutional Neural Networks (RCNNs) are a joining of Convolutional Neural Networks (CNNs), which are well-suited for feature extraction from images or spatial data, and Recurrent Neural Networks (RNNs), which are well-suited to the analysis and modelling of temporal data, and are therefore often applied to time series spatial data such as the spatial case histories associated with disease outbreaks of interest here. Indolia et al. [18] gives an introduction to CNNs. Here the recurrent part of the network is implemented using Long Short-Term Memory (LSTM) cells. These have gated memory and have grown in popularity since being introduced by Hochreiter and Schmidhuber [19] to mitigate the problems of vanishing or blowing up of error gradients often suffered with traditional RCNNs. Future details of LTSM are given by, for example, Absar et al. [20], Islam er al. [21], Tsironi et al. [22] and Yu et al. [23]. Fig 3 gives an overview of the overall neural network architecture used in the current work. The input $X_1$ is first passed through a series of time distributed convolutional filters, before being flattened and passed through an LSTM layer. Input $X_2$ is passed through a dense layer, and $X_3$ is passed through a series of convolutional filters. The outputs of each of these are then concatenated, before being passed through a dense layer and a final output layer. The output of the model, $\hat{\Theta}$, is then the predicted value of the parameter set used in the forward model, after the transformations described above are reversed.

The network is trained for 40 epochs on 7500 training observations and 100 validation observations. We use an Adam optimiser combined with a Mean Squared Error loss function. The model with the lowest loss on the validation data is saved.

The model was built using Tensorflow (version 2.4.1) on Python (version 3.8). Training on an Nvidia K-80 GPU takes around 6 hours.

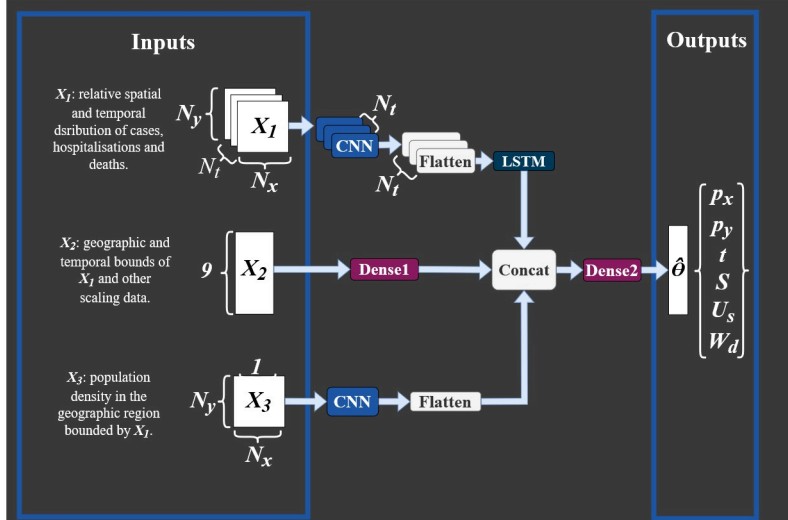

**Fig 3. The structure of the RCNN model.** Each CNN layer consists of three pairs of two-dimensional convolutional layers with 16, 32 and 64 filters respectively and a kernel size of three. Reticulated linear unit (ReLU) activation functions are used for each layer. Between each pair of convolutional layers there are two-dimensional max-pooling layers with stride two and pool size (2,2). The single LSTM layer has 256 units and uses a tanh activation function. The first dense layer (labeled Dense1) has 256 units, with a dropout rate of 0.4 and a ReLU activation function. The second dense layer (Dense2) also has 256 units, a ReLU activation function and a dropout rate of 0.2. The Output layer outputs six parameters, and uses a linear activation function.

## Results

In order to evaluate the three methods we test their ability to predict the parameter set, $\Theta = \{p_x, p_y, t, S, U_s, W_d\}$, used to produce ten simulated outbreaks. The coefficient of determination or $R^2$ score for each method across each parameter set is given in Table 4. When applied to non-linear regression, as here, the $R^2$ score can be negative. This occurs when the fit is in fact worse than the constant fit of the predicted value being taken as the mean of the training values. A positive value indicates that the fit is better than this constant fit with a maximum score value of 1 representing a prefect fit. For further details see for example Chicco et al. [24]. The ten test outbreaks represent a broad range of parameter values, and include outbreaks of between 15 and 5272 cases. For the RCNN method we additionally consider a wider test set of 700 simulated outbreaks to ensure the model has not over-fitted to the training data. This is not possible for the grid-search and MCMC approaches due to the longer time taken for each prediction.

In section C of S1 Supporting Information we use the predicted parameterisations to produce estimated spatial dose distributions using the forward model. In each case, we compare the predicted dose distribution with the distribution produced from the true parameter set. As with the results above, both the MCMC and RCNN methods perform considerably better than the grid-search approach.

**Table 4. $R^2$ scores for each method across the six source parameters.**

| Method | $p_x$ | $p_y$ | $t$ | $S$ | $U_s$ | $W_d$ |
|---|---|---|---|---|---|---|
| Grid-search | −8.4 | −16.04 | 0.93 | −0.79 | −0.85 | −0.60 |
| MCMC | 0.89 | 0.04 | 0.95 | −0.41 | −1.94 | −0.99 |
| RCNN | 0.98 | 0.98 | 0.98 | 0.68 | 0.19 | 1.00 |

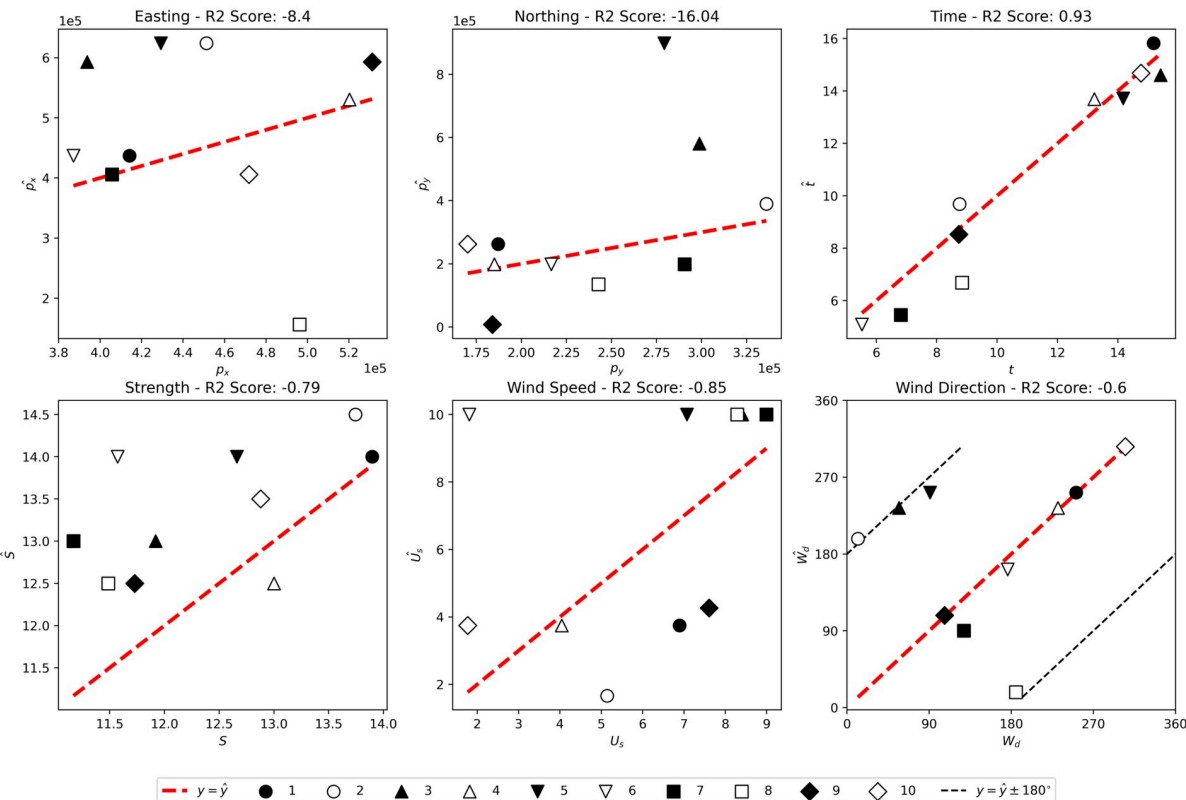

**Fig 4. Predictions ($\hat{\Theta}$) plotted against true values ($\Theta$) for the grid-search method on the ten test outbreaks.** The different symbols represent the different test simulated outbreaks, the red dashed line is the predicted value equals true value line and in the last plot (wind direction), the black dashed line shows the line on which the predicted value differs from the true value by ±180˚.

### Grid-Search

Fig 4 shows the predictions made by the grid-search method, $\hat{\Theta} = \{\hat{p}_x, \hat{p}_y, \hat{t}, \hat{S}, \hat{U}_s, \hat{W}_d\}$, plotted against the true values, $\Theta$. Different simulated outbreaks use consistent markers across each plot. The method is good at estimating the date of release, achieving an $R^2$ score of 0.93, and occasionally good at predicting the location of release. The grid-search approach often predicts the correct wind direction, and when it is wrong the error is consistently around 180˚. Notably, the method fails to accurately predict source location in predictions where $W_d$ has also been predicted to be 180˚ from the true value (predictions 2, 3, 5 and 8).

### MCMC

The predictions made by the MCMC method on the ten simulated outbreaks are shown in Fig 5. The MCMC method is more consistent than the grid-search, and provides more accurate predictions, as indicated by higher $R^2$ scores, across almost all parameters. However, the method is not able to predict source strength or wind speed with any accuracy, having negative $R^2$ scores for these parameters. The MCMC method also fails to accurately predict the location of outbreak 9, however all other outbreaks are well located.

### RCNN

The fit of a trained RCNN model to 700 test observations is shown in Fig 6 and to the same ten simulated outbeaks used to test the grid-search and MCMC methods in Fig 7. The goodness of

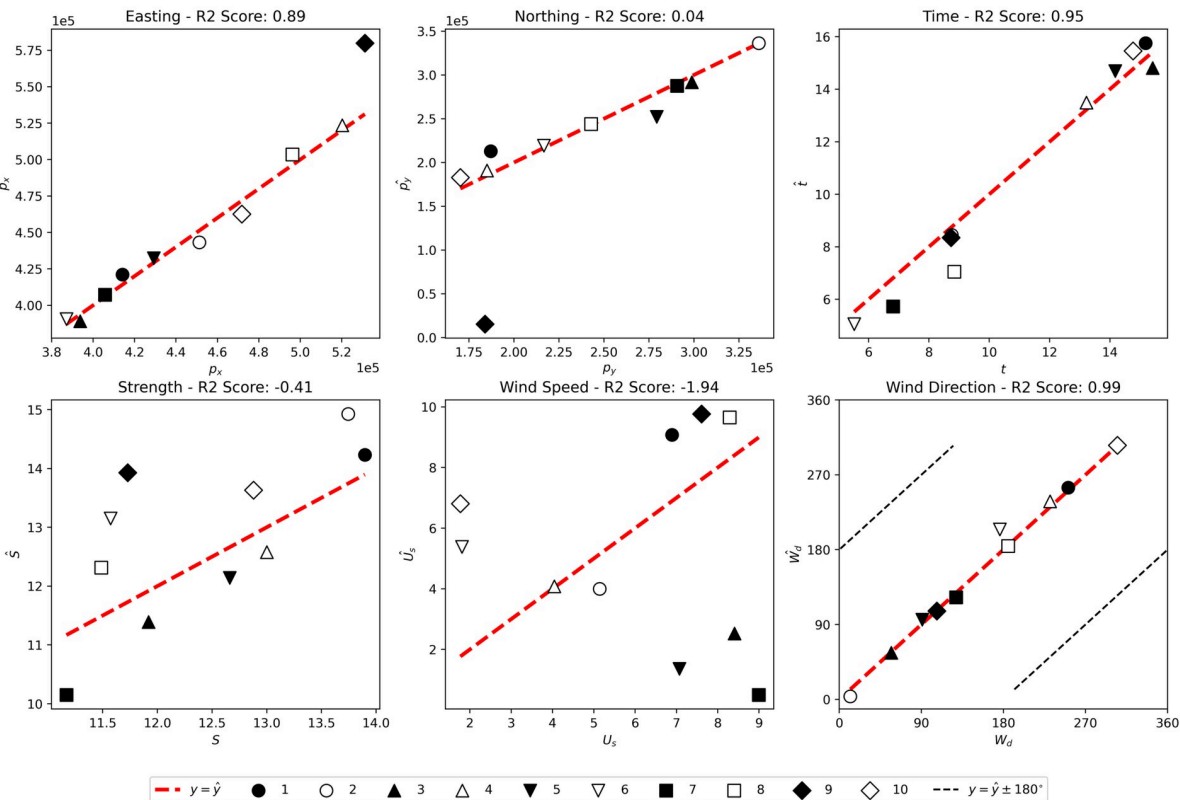

**Fig 5. Predictions ($\hat{\Theta}$) plotted against true values ($\Theta$) for the MCMC method on the ten test outbreaks.** The different symbols represent the different test simulated outbreaks, the red dashed line is the predicted value equals true value line and in the last plot (wind direction), the black dashed line shows the line on which the predicted value differs from the true value by ±180˚.

fits are comparable except for wind direction where the fit to the smaller test set is almost perfect. So the fit to the larger test set is considered.

The model is able to predict the location and timing of the release with good accuracy, achieving $R^2$ scores of 0.98, 0.98 and 0.97 on the easting, northing and time of release respectively.

The model accurately predicts the wind direction at release, $W_d$, although some predictions have an error of 180˚. Observations for which the model failed to accurately predict $W_d$ tended to have very low case numbers. The $R^2$ score on $W_d$ was 0.79, although it should be noted that this has been artificially deflated by predictions where $W_d \approx 0°$ and $\hat{W}_d \approx 360°$, or vice versa.

The model is able to predict the source strength, $S$, with a reasonable degree of accuracy, achieving an $R^2$ score of 0.79. As with the other methods, wind speed at source, $U_s$, is by far the worst prediction, with the model being only slightly better than random chance when predicting this parameter. It is likely that $U_s$ and $S$ have some interdependence making them more difficult to predict.

## Discussion

Of the three methods discussed in this paper it is clear that both the MCMC and RCNN methods provide the more accurate estimates of source parameters based on observed outbreaks. Both approaches achieve similar results on the test set of 10 outbreaks, while the grid-search

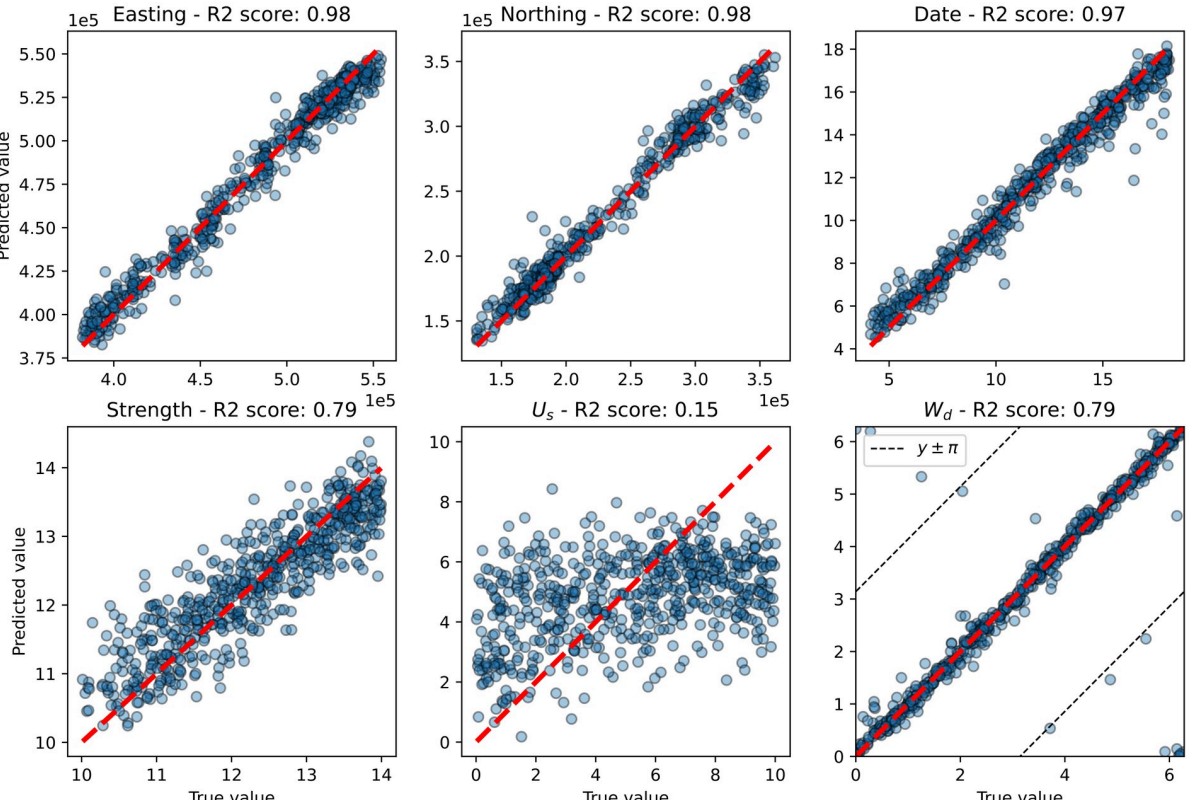

**Fig 6. Predictions ($\hat{\Theta}$) plotted against true values ($\Theta$) for the RCNN method on test set of 700 simulated outbreaks.** Each circle represents a different test simulated outbreak, the red dashed line is the predicted value equals true value line and in the last plot (wind direction), the black dashed line shows the line on which the predicted value differs from the true value by ±180°.

method consistently performs worse. Further, thanks to the fast deployment of the RCNN method, we have been able to demonstrate consistently high accuracy on much larger test sets. However, there are a number of important distinctions between the methods which may make them more or less relevant in certain situations.

The MCMC method described in this paper is relatively slow—taking a few hours to perform inference on a single outbreak. Increasing the complexity of the forward model will further increase the time taken to perform this analysis. In situations where the speed of response is critical this may be a disadvantage.

Unlike the MCMC method, the RCNN model can be quickly deployed on new case data, however the preparation and validation of synthetic training data, as well as the training of the model itself, both require considerable time investment. In instances where no suitable pre-trained model exists for a specific threat this could seriously slow the response time. As such, it is important to carefully consider which diseases most warrant the time investment required to train an appropriate RCNN model. While it is feasible to maintain a catalogue of pre-trained RCNN models to deploy in response to specific diseases, it does mean the method has limited flexibility and will not be appropriate in all circumstances. By comparison for both MCMC and grid-search methods novel diseases are handled with comparatively simple updates to the forward model with little impact on the inference time.

Both the MCMC and grid-search methods rely on an explicit likelihood function. Even with the relatively simple forward model described in this paper, the likelihood function

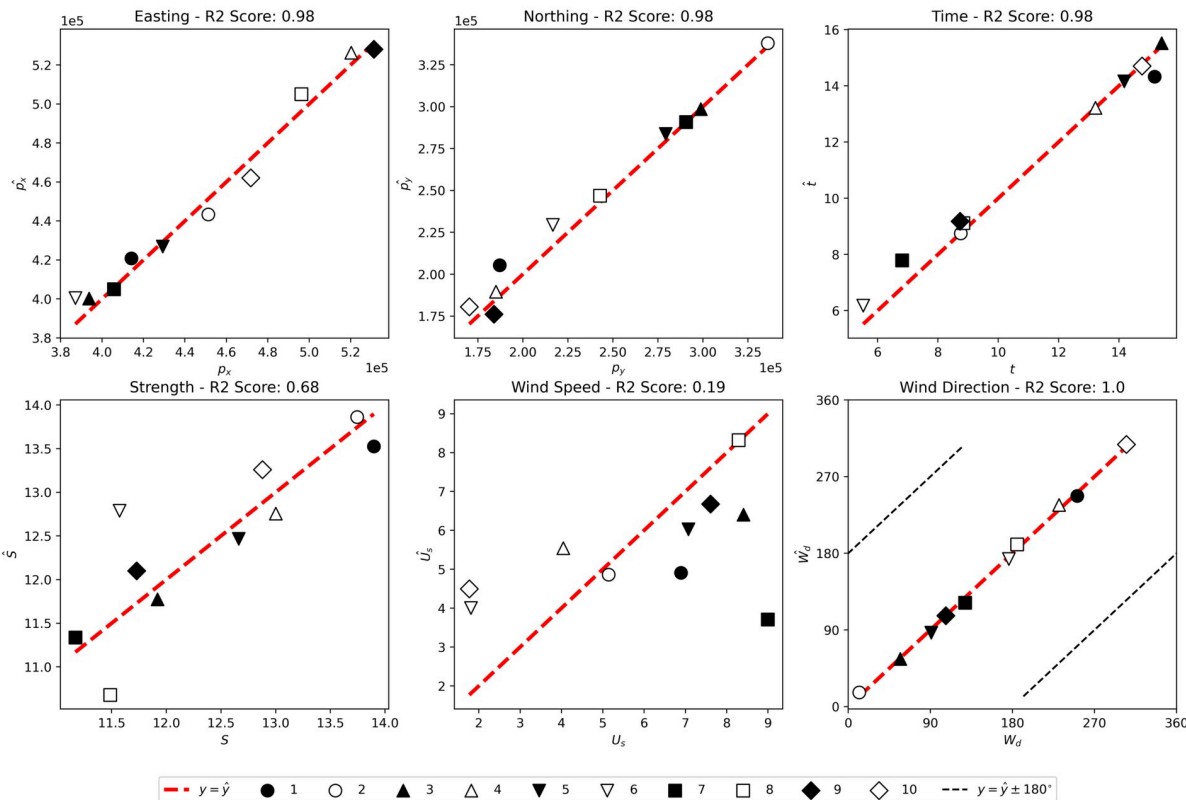

**Fig 7. Predictions ($\hat{\Theta}$) plotted against true values ($\Theta$) for the RCNN method on the ten test outbreaks.** The different symbols represent the different test simulated outbreaks, the red dashed line is the predicted value equals true value line and in the last plot (wind direction), the black dashed line shows the line on which the predicted value differs from the true value by $\pm180°$.

produces spiky distributions which can be challenging to optimise. To address this problem, we have added a number of extra steps to the MCMC approach which attempt to steer the model away from local maxima in the likelihood function, circumventing regions with negative infinity log-likelihood. As more complex forward models are introduced, which are able to more accurately simulate either the dispersion of the pathogen, the movement of susceptible populations, or the within-host dynamics of the pathogen, it will likely lead to an increasingly unstable likelihood function. This may not be a problem with the RCNN method, which does not rely explicitly on the likelihood function and is more flexible to increased model complexity—at the expense of an increased training period.

Here as simulated data is used the possible range of each parameter is know. This is clearly not the case for true data where the range used by the grid-search and MCMC methods would need to be set via expert opinion. This would be a possible source of bias and would determine the probability of finding the true global maxima, as would the choice of (uniform) priors used in the MCMC method. It is also possible that a similar bias would occur in the RCNN method if the true source parameters was outside the range used to train the model.

Other possible sources of bias include the possibility that the case data will contain sufficient information and model is misspecification. For example, could the model distinguish between a release on the edge of a populated area and one within the unpopulated area adjacent to the populated area where the cases occur? Such consideration require future research beyond the scope of this paper.

Further work is needed to demonstrate each method's ability to deal with more complex models and different disease threats. Further, it will be important to build a degree of uncertainty into the predictions produced by the models. For the grid-search and MCMC based approaches this will be relatively straight forward as the techniques lend themselves well to uncertainty quantification. For the RCNN model it may not be possible to effectively derive prediction uncertainty.

The models developed in this paper have been shown to be effective when deployed on a fully observed outbreak. It is important to note that this assumption will likely be invalid during a release event—either through cases failing to present to healthcare authorities, inaccurate or low resolution recording of symptom onset times, or through incomplete data capture processes. A natural further step in this work is to consider how well the approaches handle such incomplete or inaccurate datasets.

This paper has demonstrated three back-calculation methods for identifying source information during a deliberate release event. The novel methods—the modified MCMC and RCNN approaches—directly address two existing problems in reverse-epidemiology; that of challenging parameter-likelihood surfaces and of slow deployment. Modification of the MCMC approach suggested in [2] has allowed us to develop a method which avoids many of the problems imposed by the parameter-likelihood surface, while the RCNN approach, which does not explicitly use a likelihood function, also manages to avoid many of these problems. Further, the neural network approach allows us to undertake most of the computationally expensive inference prior to use, allowing for an inference method which can be rapidly deployed.

These results represent a significant step in source term inference and provides a basis for increased forward model fidelity in the future. For diseases such as anthrax, where effective response is dependent on the fast and efficient distribution of countermeasures where they are needed most, this ultimately reduces the burden on public health services and helps to protect more lives.

## Supporting information

**S1 Supporting Information. Supplementary material.** Section A Grid Section parameter ranges, Section B Constructing RCNN inputs, Section C Predicted dose distribution. (PDF)

## Author Contributions

**Conceptualization:** Joseph Shingleton, David Mustard, Steven Dyke, Thomas Finnie.

**Formal analysis:** Joseph Shingleton, David Mustard, Steven Dyke.

**Investigation:** Joseph Shingleton, David Mustard, Thomas Finnie.

**Methodology:** Joseph Shingleton, David Mustard, Steven Dyke.

**Project administration:** Thomas Finnie.

**Software:** Joseph Shingleton, David Mustard, Steven Dyke.

**Supervision:** Thomas Finnie.

**Validation:** Joseph Shingleton, Steven Dyke.

**Visualization:** Joseph Shingleton.

**Writing – original draft:** Joseph Shingleton, David Mustard, Steven Dyke.

**Writing – review & editing:** Joseph Shingleton, David Mustard, Steven Dyke, Hannah Williams, Emma Bennett, Thomas Finnie.

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
