## [Decision Letter · Decision Letter 0]

14 Dec 2023

Dear Dr Finnie,

Thank you very much for submitting your manuscript "Backtracking: Improved methods for identifying the source of a deliberate release of *Bacillus anthracis* from the temporal and spatial distribution of cases" for consideration at PLOS Computational Biology.

As with all papers reviewed by the journal, your manuscript was reviewed by members of the editorial board and by several independent reviewers. In light of the reviews (below this email), we would like to invite the resubmission of a significantly-revised version that takes into account the reviewers' comments.

First, apologies for the long delay in reaching this decision. We had a very hard time securing a second reviewer. As you will see, the reviews are mixed. In your response and revision, please pay particular attention to Reviewer 2's comment about providing more thorough background (literature review) and comparison to existing methods, including formal metrics for the model fit, as well as a better description of the methods in the main text. Please also make a better effort to adhere to our journal's code-sharing policy, as the reason for not sharing the code seems inadequate.

We cannot make any decision about publication until we have seen the revised manuscript and your response to the reviewers' comments. Your revised manuscript is also likely to be sent to reviewers for further evaluation.

Sincerely,

Virginia E. Pitzer, Sc.D.

Section Editor

PLOS Computational Biology

Virginia Pitzer

Section Editor

PLOS Computational Biology

First, apologies for the long delay in reaching this decision. We had a very hard time securing a second reviewer. As you will see, the reviews are mixed. In your response and revision, please pay particular attention to Reviewer 2's comment about providing more thorough background (literature review) and comparison to existing methods, including formal metrics for the model fit, as well as a better description of the methods in the main text. Please also make a better effort to adhere to our journal's code-sharing policy, as the reason for not sharing the code seems inadequate.

Reviewer's Responses to Questions

**Comments to the Authors:**

Reviewer #1: This article describes an alternative model calibration method than normally used in literature. As such it is well considered, timely and of interest.

I have the following comments prior to publication:

1) Line 22 - 24, 'two planar directions' reads oddly and planar later, perhaps 'across 2 dimensional plan is better'. Though the briggs model I think allows height as it is a Gaussian puff model with empirical diffusion terms.

2) Line 30 'realistic' may be too strong, it is certainly credible or plausible but the simplicity of Briggs given turbulent flow and population variation, with a lack of observational data of such an event means the simulation proposed is unvalidated. This does not undermine approach or results but may inflate readers expectations.

3) In intro the citation for within host model is [7] but in methods it becomes [8] can authors clarify.

4) Line 71 - reads oddly. "hospitalisations or deaths are recorded after the censor date, even if we know that" i think should be "hospitalisations or deaths are used after the censor date, even if we simulate that"

5) Line 75 - typo, no need for 'of'

6) Line 135 - " guaranteed not" is too strong, I think should be "not guaranteed" was intended, unless the authors have a citation for a proof.

7) Figures 4, 5, 6 - in cation can the authors explain the symbols and lines.

8) Line 292 - "the preparation of synthetic training data," please add something like "and validity of simulation model" as method would be prone to structural misspecification.

9) Line 311-12 - I agree but for grid search and MCMC the range of search/prior distribution definition may be considered given the structural assumptions in model to limit the range and chance of missing global maxima. This is going to potentially bias the results if the model is misspecified of course. Given the spatial pattern in Briggs is diffusion dominated after some distance most information is in the close cases - which if release was far from population centre may mean there is limited info in the data or likelihood may infer as may be equivalent to a far release. So work effort may be on the structure of the underlying model too.

Reviewer #2: The authors set out to improve upon reverse-epidemiological methods using a recurrent convolutional neural network approach. My comments are as follows.

1. This paper has no literature review. As a result, it is completely impossible to assess whether or not what the authors are contributing is a valuable contribution. For example, I did a quick google scholar search with the following search terms: “neural network” and “outbreak source location”, and saw a wide variety of papers. The geographic profiling literature has no doubt made contributions worth discussing as well. Additionally, the authors dedicate considerable time to discussing limitations to MCMC-based procedures, but don’t review them meaningfully. There is an entire body of statistical literature that addresses the same limitations to metropolis-hastings style algorithms that the authors note, but seemingly fail to incorporate in their work. Basically, the authors are trying to argue that they’ve developed a novel tool that out-performs existing methods. However, by not providing a meaningful literature review, they cannot convince this reviewer that their contribution is novel.

2. In the section introducing forward models (and the Briggs model, specifically), the authors need to introduce this better in the main text instead of the appendix. I do not feel it is sufficient for future readers of this paper to simply say that they use the Briggs’ model and then point to the appendix. I think the first page of the supplement would be better suited in the main text.

3. The RCNN is poorly described. For example, what’s an LSTM layer? It’s never described.

4. Results table 1. If these are R^2 values, why are some of them negative?

5. The entire results section is written as captions for figures and tables. Also, what are the criteria for a method being “good” or not having “a great deal of accuracy”?

6. I simply don’t understand the notion that releasing the code to this model is too much of security threat. In part this is because the authors have failed to convince me that they’ve made a novel contribution (see point 1 above). Do the authors think that if people who are prone to releasing anthrax in the UK had the code that detailed this modeling approach, they would alter their plans in any way? Not being an expert in this area, I assume their concerns are more focused on logistics than machine learning.

**Have the authors made all data and (if applicable) computational code underlying the findings in their manuscript fully available?**

Reviewer #1: **No: **They caveat this in submission. This is for editiorial team to judge on precedent rather than reveiewer.

Reviewer #2: **No: **No. This is a simulation study though.

PLOS authors have the option to publish the peer review history of their article (what does this mean?). If published, this will include your full peer review and any attached files.

Reviewer #1: No

Reviewer #2: No
---

## [Decision Letter · Decision Letter 1]

3 Jun 2024

Dear Dr Finnie,

Thank you very much for submitting your manuscript "Backtracking: Improved methods for identifying the source of a deliberate release of Bacillus anthracis from the temporal and spatial distribution of cases" for consideration at PLOS Computational Biology. As with all papers reviewed by the journal, your manuscript was reviewed by members of the editorial board and by several independent reviewers. The reviewers appreciated the attention to an important topic. Based on the reviews, we are likely to accept this manuscript for publication, providing that you modify the manuscript according to the review recommendations.

While the reviewer and editor find the revisions to be responsive to the issues raised in the first round of reviews, we still have concerns over the code availability, which we take very seriously at PLOS Computational Biology. After discussion with the Editors-in-Chief, we feel that providing no means of accessing the code is unacceptable. While we understand that the code cannot be made publicly available due to security concerns, for purposes of reproducibility and for purposes of utility, some mechanism of accessing the code should be provided, even if that requires a high bar such as approval/clearance by the UK government, NDAs/code control with the agency, etc. Alternatively, please consider whether it would be possible to provide a redacted and/or simplified version of the code based on simulated data that would not pose a security concern.

To address these concerns, please add a "Data and Code Availability" section at the end of the Methods that describes the reasons why the data/code cannot be shared publicly and includes all necessary contact information where an interested reader would need to apply in order to obtain the code.

Sincerely,

Virginia E. Pitzer, Sc.D.

Section Editor

PLOS Computational Biology

Virginia Pitzer

Section Editor

PLOS Computational Biology

While the reviewer and editor find the revisions to be responsive to the issues raised in the first round of reviews, we still have concerns over the code availability, which we take very seriously at PLOS Computational Biology. After discussion with the Editors-in-Chief, we feel that providing no means of accessing the code is unacceptable. While we understand that the code cannot be made publicly available due to security concerns, for purposes of reproducibility and for purposes of utility, some mechanism of accessing the code should be provided, even if that requires a high bar such as approval/clearance by the UK government, NDAs/code control with the agency, etc. Alternatively, please consider whether it would be possible to provide a redacted and/or simplified version of the code based on simulated data that would not pose a security concern.

To address these concerns, please add a "Data and Code Availability" section at the end of the Methods that describes the reasons why the data/code cannot be shared publicly and includes all necessary contact information where an interested reader would need to apply in order to obtain the code.

Reviewer's Responses to Questions

**Comments to the Authors:**

Reviewer #1: The authors have attended to all my comments and questions sufficiently.

**Have the authors made all data and (if applicable) computational code underlying the findings in their manuscript fully available?**

Reviewer #1: Yes

PLOS authors have the option to publish the peer review history of their article (what does this mean?). If published, this will include your full peer review and any attached files.

Reviewer #1: No

Figure Files:

Data Requirements:

Reproducibility:

References:

---

## [Editor Report · Decision Letter 2]

19 Aug 2024

Dear Dr Finnie,

We are pleased to inform you that your manuscript 'Backtracking: Improved methods for identifying the source of a deliberate release of Bacillus anthracis from the temporal and spatial distribution of cases' has been provisionally accepted for publication in PLOS Computational Biology.

Best regards,

Virginia E. Pitzer, Sc.D.

Section Editor

PLOS Computational Biology

Virginia Pitzer

Section Editor

PLOS Computational Biology

---

## [Editor Report · Acceptance letter]

31 Aug 2024

PCOMPBIOL-D-22-01837R2 

Backtracking: Improved methods for identifying the source of a deliberate release of Bacillus anthracis from the temporal and spatial distribution of cases

Dear Dr Finnie,

I am pleased to inform you that your manuscript has been formally accepted for publication in PLOS Computational Biology. Your manuscript is now with our production department and you will be notified of the publication date in due course.

With kind regards,

Zsofia Freund
